# Unlocking the Potential of Chemically Modified Carbon Gels in Gallic Acid Adsorption

**DOI:** 10.3390/gels10020123

**Published:** 2024-02-02

**Authors:** Regina C. Carvalho, Carlos J. Durán-Valle, Marta Adame-Pereira

**Affiliations:** 1Departamento de Química Orgánica e Inorgánica, Universidad de Extremadura, Avda. de Elvas, s/n, 06006 Badajoz, Spain; reginacelia_carvalho@hotmail.com (R.C.C.); martaap@unex.es (M.A.-P.); 2IACYS, Universidad de Extremadura, Avda. de Elvas, s/n, 06006 Badajoz, Spain

**Keywords:** carbon xerogel, water remediation, adsorption, gallic acid

## Abstract

This study deals with the preparation of adsorbents from a commercial xerogel by chemically modifying its surface with concentrated mineral acids and alkali metal chlorides, their physicochemical characterization, and their use as adsorbents for gallic acid in aqueous solution. Although there are publications on the use of carbon xerogels as adsorbents, we propose and study simple modifications that can change their chemical properties and, therefore, their performance as adsorbents. The adsorbate of choice is gallic acid and, to our knowledge, there is no history of its adsorption with carbon xerogels. The prepared adsorbents have a high specific surface area (347–563 m^2^ g^−1^), better pore development for samples treated with alkali metal chlorides than with mineral acids, and are more acidic than the initial xerogel (p.z.c range 2.49–6.87 vs. 7.20). The adsorption equilibrium is reached in <16 h with a kinetic constant between 0.018 and 0.035 h^−1^ for the pseudo-second-order model. The adsorption capacity, according to the Langmuir model, reaches 62.89 to 83.33 mg g^−1^. The adsorption properties of the commercial xerogel improved over a wide range of pH values and temperatures. The experimental results indicate that the adsorption process is thermodynamically favored.

## 1. Introduction

Gallic acid (3,4,5-trihydroxybenzoic acid, GA) is a molecule of interest in the pharmaceutical industry as a component of various therapeutic agents (antioxidants, anticancer, antiviral, and more) and as a preservative in food technology [1]. Moreover, GA is one of the main phytic substances present in the environment and can be considered a component of natural organic matter, which originates from the decomposition of plant biomass [2]. This pollutant is slightly toxic at low concentrations, and its accumulation in the environment should be avoided [3], especially in waters with very low biodegradability [4]. Various technologies have been used for water purification. These treatment technologies include biological methods, coagulation, nanofiltration, and adsorption. Among these techniques, adsorption is the most widely used method to remove contaminants from water with high efficiency and simple operating conditions [1,2,5,6,7].

Activated carbons [8,9] and carbon blacks [10,11] are carbon materials traditionally used in adsorption and catalysis. One of the drawbacks of activated carbons is that they are essentially microporous materials, advantageous for adsorption, which occurs mainly in the micropores. In contrast, mesoporous materials would benefit catalysis by minimizing diffusion limitations and catalyst deactivation [12,13]. In recent decades, a new carbonaceous material has been developed with exciting properties for catalytic applications and can also be used as an adsorbent.

Carbon gels (CGs) are synthetic porous carbon materials that have received attention recently due to their great potential and versatility. CGs are obtained after carbonizing organic gels prepared by polycondensation of resorcinol and formaldehyde using acid or basic catalysts [14]. Depending on the drying method of the organic gels: supercritical drying, freeze drying, and evaporative drying, the carbon gels are given different names: aerogels, cryogels, and xerogels, respectively. Evaporative drying is the most straightforward, fastest, and economical method [15,16]. The main advantage of these materials is the possibility of adapting their surface chemistry and textural characteristics and the various forms they can take (monoliths, films, microspheres, and more) for specific applications, depending highly on the synthesis and preparation conditions [14,15,16,17]. These applications include catalysis [17,18], adsorption [19,20], adsorption–photodegradation [21], and energy storage [22]. The ability to control pore size in CGs is undoubtedly an advantage compared to classic activated carbons, whose textural parameters are mainly given by the raw material used in their preparation [13]. Among the most essential and valuable properties are high porosity, surface area (400–1200 m^2^ g^−1^), and size pore distribution [23]. Job et al., 2004 and 2006, indicate that the porosity formed during synthesis hardly alters during pyrolysis, but the pH of the precursor solutions regulates these textural properties [13,24]. A chemical or physical activation process can further develop the microporosity of carbon gels [25,26]. Physical activation is carried out with a gas flow (CO_2_, N_2_, steam, or air) at high temperatures depending on the reactivity of the gas. Chemical activation involves the use of an activating agent, of which there is a greater variety. Some of the most used are alkalis such as KOH [27] or acids such as H_3_PO_4_ [28], but they can also be salts such as ZnCl_2_ [29]. In general, a higher porous development is usually obtained with chemical activation, but also a higher ash content. It is also possible to use both methods as the existing atmosphere can change the reactivity of the activating agent [30].

In this work, carbon xerogels were prepared by modifying a mesoporous carbon surface chemically with concentrated mineral acids and the adsorption of alkali metal chlorides. These treatments were carried out in our laboratory to obtain acid catalysts [31] or basic catalysts [32,33] on activated carbons. However, we have not tested them on xerogels often, but, when we have done so, we have used them as catalysts [34]. Therefore, we have considered using them as adsorbents and studied the changes in their porous texture and surface chemistry and the influence of these changes on their adsorption capacity. As an application, their behaviors as adsorbents of gallic acid from water are studied. The gallic acid adsorption process is examined from the kinetic and equilibrium point of view.

## 2. Results and Discussion

### 2.1. Chemical Characterization of Adsorbents

#### 2.1.1. Composition

The data obtained from elemental analysis (Table 1) indicate that, in general, the prepared materials have a high carbon content, followed by a high oxygen content. The oxygen content is observed to be similar when the samples are prepared with NaCl and KCl salts and increases notably for the samples prepared with acids. This increase was expected, considering the oxidizing capacity of the acids, especially nitric acid. On the other hand, the null content of N and S in the xerogels is worth noting, except for samples C-X-N and C-X-S, where the nitrogen and sulfur content increased, respectively. This fact was to be expected according to the sample preparation process. In addition, the formation of nitrogen and sulfur functional groups in C-X-N and C-X-S can be assumed. This behavior has also been observed in the microporous Norit RX3 Extra and Merck carbons, which have undergone the same treatment as nitric and sulfuric acid [31,35]. To obtain the amount of Na and K that could be present in the C-X-Na and C-X-K xerogels, WDXRF spectra were also carried out. About 1% chlorine and trace amounts of Fe and Cu were found.

The composition was measured with EDX (Appendix A), and the results are similar to Table 1. The materials are rich in carbon, only sulfur is present in C-X-S, and the most oxidized material is the C-X-N adsorbent. Neither sodium nor potassium was detected by this technique. After gallic acid adsorption, small variations in composition are observed. This method has to be considered semi-quantitative and, therefore, does not have the same precision as the results shown in Table 1.

The FT-IR spectra recorded for the xerogels studied are plotted in Figure 1. The spectra display different absorption bands whose spectral features are tentatively assigned according to the literature [36,37], as shown in Appendix A.

The band at 3400 cm^−1^ is more intense in C-X-N than in the other samples, probably due to the oxidation produced by nitric acid. For C-X-S, the band is less intense due to the possible elimination of functional groups with sulfuric acid and the lower oxidizing power of the latter, which means that not as many oxygenated functional groups originate as in the case of nitric acid. Regarding the C-X-Na sample, this band has a lower intensity because the Na^+^ cation will preferentially bind to the more electronegative atoms, i.e., oxygen, thus eliminating O-H bonds and transforming them into O-metal bonds. The ν(C=O) stress band between 1770 and 1650 is only significant in C-X-N since it is the most oxidized material. The ν(C=C) band at 1700–1500 cm^−1^ appears instead in all spectra and is somewhat more intense in C-X-N because the bonds’ symmetric vibrations do not give rise to photon absorption or emission in the infrared spectrum. However, adding heteroatoms to a carbon (oxygen in this case) breaks this symmetry and increases the number of bonds whose vibration gives rise to photon absorption. The next region of the spectrum, up to 1330 cm^−1^, corresponds to hydrogen bond bending vibrations. This zone is similar for all samples, and only the band at 1384 cm^−1^ stands out, which can be assigned to the O-H bonds of carboxyl or hydroxyl groups and the C-H bonds of olefins and methyl groups. Between 1300 and 1000 cm^−1^, another intense band appears, corresponding to the C-O bonds of both ethers and hydroxyl groups. The maximum of this band can be shifted at different wave numbers, which is related to the different ratios between oxygen atoms bonded to substituted (higher wave number) or unsubstituted (lower wave number) carbon atoms. In the C-X-N sample, there are two bands at 1550 and 1350 cm^−1^, both as shoulders of other bands and not observed in the other samples. These bands can be assigned to the presence of nitro groups (-NO_2_) resulting from the treatment with nitric acid. However, there are still different opinions about forming these functional groups with this reaction [31]. The C-X-S sample has a series of bands of small intensity at ≈1000, 700, and 600 cm^−1^ corresponding to the sulfonic groups.

The spectrum of one of the adsorbents after gallic acid adsorption was also performed. The comparative spectra are shown in Figure 2.

In the above figure, it is observed that there is an increase in the intensity of the bands after GA adsorption. Some of the most representative bands of the GA are the centered at 3420 cm^−1^, 1617 cm^−1^, and 1380 cm^−1^. According to the literature [38,39,40], the broad and strong bands located between 3650 and 3200 cm^−1^ and at 1617 cm^−1^ that also appear in the FT-IR spectrum of pure gallic acid are attributable to the stretching vibrations of the OH groups and the C-C bonds of the aromatic rings of GA. The band at 1380 cm^−1^ can be assigned to the O-H bonds of carboxyl or hydroxyl groups and the C-H bonds of methyl groups.

Table 2 shows the elemental analysis results (mass composition) measured by the XPS technique. In general, the variations in composition are similar to those shown in Table 1. The oxygen and sulfur content increases for sample C-X-S, and, for sample C-X-N, the oxygen and nitrogen content increases. In addition, a slight decrease in oxygen content is observed for xerogels treated with alkali metal chlorides. In general, no significant changes are observed, so it can be assumed that the modification caused by the chemical treatment is not limited to the xerogel surface. Comparing the results of the global elemental analysis (Table 1) with the XPS results (Table 2), which study a shallower area, a C enrichment and a decrease in the O content at the surface are observed.

The XPS spectra of C 1s and O 1s are plotted in Appendix A, and the deconvolution results for the C1s, O1s, N1s, and S1s peaks are shown in Table 3, Table 4, Table 5 and Table 6.

The C1s orbital has peaks with a maximum near 284.8 eV. The components of these peaks originate in the more or less oxidized forms of the element carbon. The assignment of these components to the different chemical structures were carried out according to what was described in the literature [31,41,42,43,44,45,46,47,48]. The peak around 284.8 eV corresponds to the C-C bonds, aromatic bonds of the carbon basal planes and aliphatic hydrocarbons, and C-H bonds. The second peak around 286.0 eV corresponds to partially oxidized C, as hydroxyl or carbonyl groups. The peaks near 289.0 eV correspond to esters, acids, anhydrides, or amides, which are functional groups in which the carbon element is bonded to two more electronegative atoms. According to Table 3, the main component is graphitic carbon. The treatment with nitric acid (C-X-N) increases the amount of highly oxygenated functional groups since it increases the intensity of the peak at 289 eV. The intensity of the peak at 286 eV decreases with all treatments, which can be explained by the fact that aldehydes and ketones are usually more reactive functional groups than others, such as alcohols or esters. The treatment with KCl results in a considerable reduction in the surface area of the xerogel as an increase of the 284.8 eV component and a decrease of the other two components is observed.

The interpretation of O 1s XPS spectra in complex materials such as carbon xerogels or other carbonized materials is complex as one functional group can provide more than one peak and very different functional groups can provide very close signals. Therefore, we have preferred not to make any interpretations. However, it should be noted that no signal is observed near 530–531 eV, which indicates that there are no oxygen compounds with alkali metals in the C-X-Na and C-X-K samples.

As with nitrogen, sulfur was only detected in one carbon, the C-X-S sample. The observed doublet is due to the presence of sulfur in a high oxidation state (SO_3_H, -O-SO_3_H), which would agree with the sulfonation of carbon [31,42,45]. The S 2p spectrum is shown in Figure 3.

Regarding the analysis of the N1s orbital, only nitrogen was found, as expected, in the C-X-N sample. The first component presents an energy value of near 401.0 eV due to nitrogen in reduced form, i.e., as pyrolytic or pyridinic nitrogen [44,45] or amine [46] because of HNO_3_ reduction. Other peak component was detected at a higher value (405.9 eV), indicating that nitrogen is bound to oxygen [44,45]. These results may be due to traces of adsorbed HNO_3_, which are by-products of HNO_3_ partial reduction or nitration. The N 1s spectrum is shown in Figure 4.

#### 2.1.2. Acidic and Basic Properties

The p.z.c. values and the amount of acidic and basic groups (Table 7) highly depend on the xerogel treatment.

In general, activated carbons with high oxygen contents have low p.z.c. values and vice versa. This is related to the oxidation process of the carbons, which usually gives rise to acidic functional groups. The oxidation of the xerogels produces a noticeable change in the acid/base character for C-X-S and C-X-N relative to C-X, which is the starting material. These p.z.c. values are consistent with the oxygen content of the samples; C-X has a content of 12.0% (Table 1), and C-X-S and C-X-N have a content of 15.5% and 21.8%, respectively; as expected, C-X has a higher pH_pzc_ value, i.e., it is more basic than C-X-S and C-X-N. These changes indicate that most oxygenated functional groups produced after C-X oxidation are acidic. Also, the formation of sulfonic groups on C-X-S, which are more acidic than the carboxylic acid functional groups, should be noted. On the other hand, no significant changes are observed in the p.z.c. of carbon treated with alkali metal salts. This treatment has been described to increase the basicity of other carbons, mainly if a potassium salt is used [33].

In relation to the acidic and basic groups, the treatment with H_2_SO_4_ and HNO_3_ does not modify the number of basic groups but significantly increases the number of acidic groups. The treatment with NaCl and KCl increases both groups, although not in a significant proportion. The relationship between p.z.c. and the number of acidic and basic groups is not direct; as shown in Table 7, the treatment with HNO_3_ results in more acidic functional groups than the treatment with sulfuric acid but not in a more acidic p.z.c. This is because, in addition to the number of acidic or basic groups, the strength of acids or bases of the functional groups must be considered. These results agree with the existence of strongly acidic sulfonic groups in C-X-S.

### 2.2. Structural Characterization

#### 2.2.1. N_2_ Adsorption

The N_2_ adsorption isotherms at −196 °C obtained for the adsorbents studied are shown in Figure 5. These isotherms present a hysteresis cycle wider in C-X-N than in the rest of the samples, which allows them to be classified as type IV isotherms of the classification system proposed by BDDT [49] and the most recent IUPAC technical report [50], indicating that they are mesoporous adsorbents. The hysteresis cycle is of type H4, as the adsorption branch is intermediate between isotherm types I and II. In addition, the fact that the cycle closes at a relative pressure close to unity indicates that the pores are slot-shaped. 

The isotherms and textural data (Table 8) show that the modifications made to the activated carbons cause a moderate change in the specific surface area (S_BET_). The treatment with mineral acids (C-X-S and C-X-N) resulted in carbons with a lower S_BET_. In contrast, the treatment with alkali metal salts (C-X-Na and C-X-K) slightly increased the BET surface area. Regarding the volume of micropores (Dubinin–Astakhov, DA; and Dubinin–Radushkevish, DR), the data show a decrease in the volume of micropores in C-X-S and C-X-N, being more pronounced in the latter, and an increase in C-X-Na and C-X-K. These results are consistent with those obtained for S_BET_ since the specific surface area depends mainly on the number of micropores. With regard to the mean pore diameter (measured by the DR method) of the samples, it varies from larger to smaller according to C-X-N > C-X-S > C-X-Na > C-X > C-X-K. This value would also be due to the fact that in the samples prepared with salts, there are more narrow pores and, therefore, more specific surface area; in the acid-treated samples, the opposite effect occurs.

#### 2.2.2. Mercury Porosimetry

Figure 6 shows the cumulative pore volume obtained for the carbons. Treating C-X with mineral acids and alkali metal salts causes significant variations in wide porosity (Table 9). C-X treatment with mineral acids shows a slight decrease in mesopore volume and a more pronounced decrease in macropore volume, which is very similar for C-X-S and C-X-N. In treating C-X with alkali metal salts, there is a significant increase in the volume of meso- and macropores for the C-X-Na and C-X-K samples, being much higher for C-X-Na. On the other hand, C-X shows a monomodal porosity distribution, with a pore diameter between 0.01 and 0.10 µm, which is maintained in the acid treatment (samples C-X-S and C-X-N). Samples C-X-Na and C-X-K show a bimodal distribution since new pores with a pore diameter between 1 and 10 µm appear via the treatment with alkali metal salts (Figure 7).

#### 2.2.3. Scanning Electron Microscopy

Some representative images of the carbons obtained via SEM are shown in Appendix A. The micrographs show that C-X-S and C-X-N (Appendix A) have a similar structural appearance to commercial carbon xerogel. However, the C-X-Na and C-X-K samples have a different appearance, with a smaller particle size and higher surface roughness. This cannot be explained by the mechanical stirring time as the acid-treated samples were stirred for one week for the cleaning process. The samples treated with salts were only agitated for 90 min. Temperature is not likely to play a role, as carbonaceous materials are usually very temperature-stable in the absence of oxidizing agents in the medium. Therefore, it must be assumed that the sodium and potassium salts affect and modify the structure somehow, and we should consider whether there is any catalytic effect in the future.

### 2.3. Gallic Acid Adsorption

#### 2.3.1. Adsorption Kinetics

The data of C (mg L^−1^) as a function of t (h) obtained for the adsorption systems formed by the GA in an aqueous solution and the adsorbents are plotted in Figure 8. From Figure 8, it can be inferred that a good part of the GA in the solution is adsorbed at contact times of ≤16 h by the adsorbents. It should be noted that the amount of GA adsorbed concerning the initial concentration was similar for all adsorbents except for C-X-N. This result agrees with the development of porosity in the samples. Table 8 shows that V_DR_ is higher for samples treated with alkali metal salts than those treated with mineral acids. On the other hand, the slight variations obtained in the adsorption of GA by the adsorbents could be due to the surface heterogeneity of the samples.

The fit of the kinetic data to the pseudo-first- and pseudo-second-order models is shown in Table 10. According to the R^2^ values, the kinetic data best fit the pseudo-second-order model. Moreover, the calculated value of q_e_ in the pseudo-second-order model is closer to the experimentally observed one, which supports its validity. The highest value of q_e_ corresponds to C-X, while the kinetic constant k_2_ is higher for C-X-S.

#### 2.3.2. Adsorption Isotherms

The adsorption isotherms obtained for the samples and GA in an aqueous solution are represented in Figure 9. The isotherms have a concave shape concerning the abscissa axis, which is characteristic of group L isotherms according to the Giles classification [51]. This indicates no strong solvent competition for the active adsorption centers. The results show that the oxidation of the carbons because of the acid treatment does not significantly modify the shape of the isotherms. The shape of the isotherms does change to a greater extent for carbons treated with alkali metal salts, although they still belong to the L group. Likewise, the isotherms belong to subgroups 2 or 3, which is typical of systems in which adsorption occurs by filling consecutive layers. Given Figure 9, the treatment with mineral acids and alkali metal salts improves the adsorption capacity, especially for the C-X-Na adsorbent.

Table 11 shows the values from fitting the experimental data to the different isotherm models applied. In relation to the adsorption capacity of the carbons, according to the Langmuir model, C-X-S is the highest, followed by C-X-Na and, finally, with the lower adsorption capacity, C-X carbon. For the Freundlich isotherm model, the K_F_ parameter indicates the adsorption capacity, with the samples prepared with alkali metal salts showing the highest adsorption capacity, followed by the sample prepared with nitric acid, and, finally, C-X and C-X-S. The latter data are more similar to the data obtained experimentally. Moreover, the R^2^-value is higher in the Freundlich model for almost all adsorbents. This can be explained in two ways. One is that there is a good adsorption capacity in the monolayer, but it is accompanied by adsorption in successive layers. This effect would be particularly important for C-X-S and not as important for C-X. The other possible explanation is that there is heterogeneity at the surface and that the active sites have a wide range of adsorption energies, which is more in agreement with the Freundlich model than with the Langmuir model. Further, in the case of carbonaceous materials, with a complex structure, this is to be expected. In addition, the chemical modification of the C-X carbon xerogel changes this aspect of its properties, as the starting material is the only one that fits the Langmuir model better. The parameter n indicates that the adsorption process is dominated by physisorption. The better adsorption of the C-X-Na and C-X-K samples can be explained by the presence of Na^+^ and K^+^ cations on the surface. These can cause electrostatic attraction on the gallic acid since, under the conditions under which it was made, the gallic acid must be in the form of anion gallate. Other types of interactions that may be present in the other carbon xerogels (pi–pi interactions, dipolar interactions with oxygenated groups, etc.) also exist in C-X-Na and C-X-K.

Table 12 shows some published results on the adsorption of gallic acid on different adsorbents.

The results obtained in our laboratory are superior to most published results. It is difficult to compare these data as the conditions under which they were obtained may be different. For example, the best adsorption result shown in Table 12 (the first in Table) is obtained using a 20 times higher concentration of gallic acid. In general, the use of activated carbons and polymers provides good results.

#### 2.3.3. Influence of pH

The amount of GA removed as a function of pH is shown in Figure 10. This figure shows that the medium’s pH strongly influences the adsorption capacity of GA in an aqueous solution. According to previous works [62,63], GA is a molecule easily ionized due to the deprotonation of the carboxylic group. Consequently, gallic acid can exist in neutral or anionic form, depending on the pH of the medium. Increasing the pH leads to deprotonation of the carboxylic group in the GA structure. Moreover, an increase in pH above 7 marks the formation of the dianion. Adsorption increases at a low pH. This is explained by the fact that under these conditions, gallic acid is, as such, in its neutral form, and not as gallate. Figure 11 shows the calculated electron density in gallic acid molecules and gallate anions. It can be seen that gallic acid is practically neutral with some positive charge near the protons (dark blue); however, in gallates, there are no areas of positive charge, and negative charge predominates. The adsorbents are protonated, and this positive charge attracts the gallic acid molecule, which has several dipoles and, therefore, negatively charged areas in its molecule. In the case of the C-X-Na and C-X-K samples, there is an added effect of alkali cations. In the case of the C-X-N sample, it possesses a greater number of oxygenated functional groups and, therefore, greater possibilities of bonding by Van der Waals forces. 

At a pH closer to neutral, adsorbents may have a surface charge close to zero or negative in the more acidic xerogels. This repels the gallate anion, although the attraction of Na+ and K+ cations must still be considered. At a basic pH, both the adsorbate and the adsorbents are negatively charged, which explains the almost zero adsorption capacity. Again, under these conditions, the adsorption shown by the C-X-Na and C-X-K xerogels can be explained by the presence of the metal cations. A graphical representation of the described mechanism is shown in Figure 12.

#### 2.3.4. Influence of Temperature

Figure 13 shows the amount of non-adsorbed GA (C_0_ = 50 mg L^−1^) as a function of temperature for the adsorbent samples. From the figure, it can be seen that the amount of adsorbed GA increases slightly with temperature. From the adsorption equilibrium data measured at different temperatures, the entropy, enthalpy, and Gibbs free energy of GA were calculated with different carbons (Table 13).

The enthalpy increase is positive in all cases, indicating that the process is endothermic. The entropy is positive, which is to be expected if adsorption increases with temperature. The calculation of the Gibbs free energy at 298 K yields negative values in all cases except for C-X-N, indicating that adsorption is a spontaneous process for all samples except for C-X-N, where a non-spontaneous process takes place.

#### 2.3.5. Reuse Experiment

Experiments were carried out to study the possible extraction of GA as well as the reuse of adsorbents. The C-X-Na xerogel was chosen as it had adsorbed the highest amount according to the adsorption isotherm. The adsorption of GA was carried out at room temperature and a pH of 4.0, as these conditions were high adsorption conditions according to the data shown above. Regarding desorption, two parameters were studied independently. On the one hand, a basic pH was used since the worst adsorption performance was achieved under this condition (Experiment A). On the other hand, desorption was carried out at a high temperature (95 °C) since the amount adsorbed decreases with increasing temperature (Experiment B). The results obtained are shown in Table 14 and Table 15.

The adsorption capacity decreases after the first cycle as the removal of GA is not complete. There are no major differences between the results of the second and third cycle, so it can be assumed that the surface has stabilized and the xerogel can still be used although with a somewhat lower performance than in the first cycle. The amount extracted is lower in the first cycle, both in quantity and as a percentage of the amount adsorbed in that cycle. This can be explained by the fact that the gallic acid occupies the positions where the binding is strongest in the first cycle, resulting in a partly irreversible process. In the subsequent cycles, these positions are mostly occupied by the active sites with lower binding energies, so the process is reversible.

As in the previous experiment, the adsorption capacity decreases after the first cycle. Again, there are no major differences between the results of the second and third cycle, so it can be assumed that the surface has stabilized and the xerogel can be used repeatedly, although with a somewhat lower performance than in the first cycle. The extraction method provides a lower yield than the previous one.

## 3. Conclusions

The chemical modification of an activated carbon changes the starting material’s physical structure. The treatment with mineral acids and alkali metal salts on the starting xerogel modifies its adsorptive properties. The treatment with NaCl and KCl salts results in more porous development than the treatment with H_2_SO_4_ and HNO_3_. In terms of chemical structure, using oxidizing acids increases the oxygen content and nitrogen and sulfur content when nitric or sulfuric acid are used, respectively. As a consequence of these changes, the p.z.c. of the adsorbents treated is thus significantly lower than that of the starting material. The treatment with mineral acids increases the number of acid groups and maintains the number of basic groups. On the other hand, the treatment with alkali metal salts maintains or increases the number of acid groups and markedly increases the number of basic groups.

The best adsorbents for gallic acid for each treatment are C-X-N for the acid treatment and C-X-Na for the basic treatment. In relation to the kinetics of the adsorption systems studied, the experimental data best fit the pseudo-second-order model. No theoretical mathematical model was found that adequately describes all the adsorption isotherm experiments performed. The adsorption capacity increases with a decreasing pH value and temperature of the solution. The calculated entropy, enthalpy, and Gibbs free energy values in the gallic acid adsorption experiments indicate that this process is thermodynamically favored.

## 4. Materials and Methods

### 4.1. Raw Material

This study used commercial granular carbon CX-5 (hereinafter C-X) supplied by Xerolutions S.L. (Gijón, Asturias, Spain) This material is a mesoporous-activated carbon obtained by carbonization of a polymer xerogel. The carbon was sieved using an IKA A10 basic mill, and a particle size between 1 and 2 mm was selected.

### 4.2. Synthesis of Modified Carbons

C-X carbon was used to obtain the catalysts and adsorbents. The acid catalysts and adsorbents [31] were prepared by suspending 5 g of C-X in 100 mL of H_2_SO_4_ (98%, Labkem (Premia de Dalt, Barcelona, Spain)) or HNO_3_ (65%, Fisher Scientific (Alcobendas, Madrid, Spain)). The mixtures were stirred for 90 min at room temperature. The resulting carbons were filtered and washed with distilled water for one week, changing the wash water twice daily. Yields: 96% and 102%, respectively.

The basic catalysts and adsorbents were prepared by ion exchange of the activated carbon with 1 M NaCl (>99.5%, Fisher Scientific) or 1 M KCl (>99.5%, Fisher Scientific) solution [32]. For this purpose, 5 g of activated carbon was suspended in 50 mL of salt solution (solution to carbon ratio 10:1). The samples were shaken for 48 h at 80 °C. The resulting carbons were filtered and washed to obtain a chloride-free material. Yields: 94% and 96%, respectively.

The preparation conditions and code of the catalysts/adsorbents obtained are provided in Table 16 and Figure 1.

### 4.3. Adsorbent Characterization

An elemental analysis (C, H, N, and S) was carried out using the LECO CHNS-932 equipment, LECO Corporation, St. Joseph, MI, USA, and the difference was assigned to oxygen content.

The FT-IR spectra for the samples were recorded on a Nicolet FT-IR spectrometer (Thermo Scientific, (Alcobendas, Madrid, Spain)) using a KBr disc. The discs were prepared using 10 mg of the sample and 400 mg of KBr. One pure KBr disc of the same mass as the sample disc was also prepared and its spectrum was used as background. FT-IR spectrums were recorded between 4000 and 400 cm^−1^ using 50 scans and a resolution of 0.4 cm^−1^.

X-ray photoelectron spectroscopy (XPS) of the samples was performed on a Thermo Scientific K-Alpha equipment, Thermo Scientific, equipped with a monochromatic Kα radiation source of Al at 1486.68 eV, voltage of 12 KV, and irradiation perpendicular to the sample of 90°. A Flood Gun charge compensation device was activated, and a spot dust analysis was performed.

Wavelength dispersive X-ray fluorescence (WDXRF) was performed on a Bruker S8 Tiger, Blue Scientific Limited, St. John’s Innovation Centre, Cambridge, UK, with the samples prepared as powder deposit on film, excitation with Rh X-ray source at 4 kW, maximum voltage 60 kV, and maximum intensity 170 mA.

The point of zero charge (p.z.c.) values were determined under batch conditions using the method proposed by Nabais and Carrott [64].

The amount of acid and basic groups in the different samples was determined in duplicate by acid–base titration. To determine the acid groups, 0.15 g of the sample was mixed with 30 mL of 0.01 M NaOH for 24 h at 25 °C in a thermostatic bath with continuous agitation, and 20 mL of the filtrate was titrated with 0.01 M HNO_3_. Conversely, for the determination of the basic groups, 0.15 g of the sample was mixed with 30 mL of 0.01 M HNO_3_ for 24 h at 25 °C, and 20 mL of the filtrate was titrated with 0.01 M NaOH. Phenolphthalein was used as an indicator in the acid–base titration. From the volumes obtained in the titrations, the amount (meq g^−1^) of acidic and basic sites was calculated.

The textural characterization of the samples was carried out by studying the N_2_ adsorption isotherm realized at −196 °C in a Quantachorme Autosorb-1 apparatus, previously degassing the sample at 110 °C for 24 h, and using mercury porosimetry realized in a PoreMaster 60 porosimeter, both from Quantachrome Company (Boynton Beach, FL, USA).

With a scanning electron microscope model Quanta 3D FEG (FEI Company, (Hillsboro, OR, USA)) operating in high-vacuum mode (6·10^−4^ Pa) at an accelerating voltage of 10 kV and using a secondary electron detector for high-vacuum SEM images and EDX spectra were obtained.

### 4.4. Adsorption of Gallic Acid

In the study of the adsorption process, gallic acid monohydrate (99%, Panreac) was used. Adsorption kinetics and equilibrium studies were performed using the batch procedure at 30 °C under constant stirring. For kinetics, 200 mg of adsorbent and 200 mL of a 50 mg L^−1^ aqueous solution of gallic acid (pH ≈ 4) was used. These experiments were performed in duplicate until the adsorption equilibrium was reached. The criterion for defining the equilibrium time was that the adsorption rate should be less than 1% of the initial rate. The adsorption isotherms were measured using 40 mg of adsorbent and 40 mL of adsorbate in aqueous solution of different concentrations (from 0 to 100 mg L^−1^). All gallic acid measurements were completed using UV-Vis spectrophotometry in a Shimadzu UV-1800 spectrophotometer, (Duisburg, Germany). The wavelength corresponding to the maximum absorbance was obtained at 268 nm. The adsorption capacity of gallic acid was quantified using the mass balance Equation (1) as follows:(1)$$Q_{e}=\left(\frac{C_{0}-C_{F}}{W}\right)\times V$$
where *Q_e_* is the amount of gallic acid adsorbed per gram of adsorbent (mg g^−1^); *C*_0_ and *C_F_* are the initial and final gallic acid concentrations (mg L^−1^) in the aqueous solutions, respectively; *W* is the adsorbent mass (g); and *V* is the volume of gallic acid solution (L). All adsorbents were performed in the subsequent studies of the adsorption capacity of gallic acid at different pH values (acidic, 2.2; neutral, near 7.0; alkaline, near 10.5) to investigate the influence of pH and temperatures (ranging from 35 °C to 65 °C) to investigate the influence of temperature and calculate the thermodynamic parameters of the adsorption of using 40 mg of adsorbent and 40 mL of a 50 mg L^−1^ aqueous solution of gallic acid.

The kinetic data were fitted to the pseudo-first-order (Lagergren) [65], pseudo-second-order (Ho and McKay) [66], and Weber and Morris [67] kinetic models, and isotherm data were fitted to the Langmuir [68] and Freundlich [69] models (Appendix A).

### 4.5. Reuse Experiments

Two reuse experiments of the C-X-Na adsorbent with gallic acid extraction were carried out. In experiment A, 100 mg of the xerogel and 100 mL of a solution of 100 ppm concentration at pH = 4.0 were used and stirred for 24 h at room temperature. The GA extraction was carried out with 100 mL of distilled water at pH = 11, obtaining the pH with the addition of NaOH and also for 24 h at the same temperature. The charcoal obtained, dried but not washed, was reused on two more occasions. In experiment B, adsorption was performed in the same way, but the extraction was carried out with distilled water at 95 °C. The process was repeated in the same way on two more occasions.

### 4.6. Computational Study

The computational study of the electronic density of gallic acid was performed using the Gaussian16 package at the M06-2x/6-311G++(d,p) level of theory. The calculations were carried out in water (SMD model).

## Data Availability

All data and materials are available on request from the corresponding author. The data are not publicly available due to ongoing researches using a part of the data.

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
