# Peer review of "Unlocking the Potential of Chemically Modified Carbon Gels in Gallic Acid Adsorption"

_gels, 2024, doi:10.3390/gels10020123_

Round 1

Reviewer 1 Report

Comments and Suggestions for Authors

1.    Abstract should be expanded showing the importance of the research and what is new in the study.

2.    Authors should add some examples of chemical activation step of Carbon materials and their references in the Introduction part.

3.    In the experimental part, the preparation scheme must be drawn to facilitate the preparation steps.

4.    In the result and discussion section, I suggest moving Table 3 to supplementary file.

5.    A Table of Comparison between the obtained results with the other previous results in literature must be added.

6.    Study the chemical stability of the prepared adsorbent by recycling step, also the authors must do XRD, FTIR and EDS analysis for the adsorbents before and after adsorption step of Gallic acid.

7.    The references list should be updated.

Reviewer 2 Report

Comments and Suggestions for Authors

In the study authors have prepared carbon xerogels by modifying a mesoporous carbon surface chemically with concentrated mineral acids and used them as adsorbents and studied the changes in their porous texture and surface chemistry and the influence of these changes on their adsorption capacity. As an application, its behavior as an adsorbent of gallic acid from water was studied. The gallic acid adsorption process was examined from the kinetic and equilibrium point of view. A detailed adsorbent characterization study was also carried out which adds value to the manuscript. However, the following points need to be considered while revising the manuscript.

1. Authors must include statistical data (std. deviations) in Figures 7-10. 

2. Authors need to compare and analyze the adsorption capacities of these adsorbents with the data reported in the literature. 

3. Authors need to conduct and "Desoprtion studies" to verify the reuse potential of the adsorbents used in the study. 

4. Please add more quantitative results to your abstract.   

Reviewer 3 Report

Comments and Suggestions for Authors Point 1:- Adsorption process is usually reversible or ireversible on Activated carbons gel
Point 2:- Can you explain Langmuir isotherm in these process not state only equaion
Point 3:- Can you give more explaination by strcutural view point as it gives more presentable view
Point 4:- What is compound 1 and compound 2 in figure 2 and figure 3
Point 5:- In conculsion you had mentioned about pH but not temperature
Point 6:- In table 1, you had mentioned room temperature ( give value in ranges) and 80 degree ( give in range)
Point 7:- What is the yield of
modified carbons after synthesis

Round 2

Reviewer 1 Report

Comments and Suggestions for Authors

Accept in present form.

Reviewer 2 Report

Comments and Suggestions for Authors

Authors have made necessary corrections in the manuscript.